# Inhibition of Cyclooxygenase-2 Alters Craniofacial Patterning during Early Embryonic Development of Chick

**DOI:** 10.3390/jdb9020016

**Published:** 2021-04-23

**Authors:** Bhaval Parmar, Urja Verma, Kashmira Khaire, Dhanush Danes, Suresh Balakrishnan

**Affiliations:** Department of Zoology, Faculty of Science, The Maharaja Sayajirao University of Baroda, Gujarat 390002, India; bhaval.p-zoophd@msubaroda.ac.in (B.P.); urja.verma-zoophd@msubaroda.ac.in (U.V.); kashmira.k-zoophd@msubaroda.ac.in (K.K.); dhanush.danes-zoo@msubaroda.ac.in (D.D.)

**Keywords:** cranial neural crest cells, embryogenesis, development, cell migration

## Abstract

A recent study from our lab revealed that the inhibition of cyclooxygenase-2 (COX-2) exclusively reduces the level of PGE_2_ (Prostaglandin E_2_) among prostanoids and hampers the normal development of several structures, strikingly the cranial vault, in chick embryos. In order to unearth the mechanism behind the deviant development of cranial features, the expression pattern of various factors that are known to influence cranial neural crest cell (CNCC) migration was checked in chick embryos after inhibiting COX-2 activity using etoricoxib. The compromised level of cell adhesion molecules and their upstream regulators, namely CDH1 (E-cadherin), CDH2 (N-cadherin), MSX1 (Msh homeobox 1), and TGF-β (Transforming growth factor beta), observed in the etoricoxib-treated embryos indicate that COX-2, through its downstream effector PGE_2_, regulates the expression of these factors perhaps to aid the migration of CNCCs. The histological features and levels of FoxD3 (Forkhead box D3), as well as PCNA (Proliferating cell nuclear antigen), further consolidate the role of COX-2 in the migration and survival of CNCCs in developing embryos. The results of the current study indicate that COX-2 plays a pivotal role in orchestrating craniofacial structures perhaps by modulating CNCC proliferation and migration during the embryonic development of chicks.

## 1. Introduction

Craniofacial development involves the formation of cranial neural crest cells (CNCCs) via epithelial–mesenchymal transition (EMT), induction, delamination, and migration, followed by the morphogenesis of various organs of an organism [1]. The above-mentioned events are tightly regulated by several genes that coordinate for craniofacial formation and patterning [2]. CNCCs are clusters of multipotent cells and fate-restricted progenitors that can differentiate into a multitude of tissue types based on the molecular signals they receive [3]. Their precursors undergo EMT and migrate from the forebrain, midbrain, and rhombomeres of the hindbrain to populate at the pharyngeal arches and contribute to the patterning of head and face structures. Once CNCCs pass through the EMT process, they begin migration. During migration, they proliferate and increase the pool of cells. The whole process of CNCC migration and proliferation is governed by various signaling pathways such as Fgf (Fibroblast growth factors), Wnt (Wingless-related integration site), TGF-β, and BMP (Bone morphogenetic protein) [3,4]. When the migration or differentiation of CNCCs is disrupted, defects of descendant tissues occur, which result in craniofacial malformations, the most common birth defect in humans [5].

Based on studies involving a wide array of model organisms, it can be construed that the molecular organizers of CNCC migration are conserved across various classes of vertebrates [4,6,7]. The canonical Wnt/β-catenin signaling pathway is reported to play a major role in the formation and progression of CNCCs, as it influences both delamination and migration by interacting with BMP4 and TGF-β, respectively [8,9,10,11]. Delamination is a collective effort orchestrated by downstream targets of Wnt3A and BMP4 signaling. Interestingly, BMP also plays a pivotal role in the specification of CNCCs into glial cell lineages [12]. However, the proximate regulators of TGF-β such as Snail1 (Zinc finger protein SNAI1), Twist, MSX1, MSX2, and Sox9 (SRY-related HMG-box genes) maintain the pluripotency of CNCCs during migration. Additionally, TGF-β decreases the levels of E-cadherin, with a concomitant surge in N-cadherin, by regulating the expression of Twist, Snail, and Slug, which triggers the transition of cells from epithelial to mesenchymal lineage in the neural tube [13].

Among the variety of regulatory factors reported to be expressed during delamination and migration, one key molecule is COX-2, an inducible isoform of cyclooxygenase, which catalyzes the formation of PGE_2_ from arachidonic acid [14]. COX-2-mediated PGE_2_ synthesis plays a crucial role in cellular events such as cell proliferation, migration, EMT, and differentiation by modulating a myriad of signal transduction pathways such as Wnt/β-catenin, BMP, and TGF-β [15,16,17]. A study by Jang and coworkers (2009) showed that COX-2 predominantly induces EMT in colon cancer cells by altering the expression of E-cadherin [18]. COX-2 is known to regulate the metastasis of cancerous cells by interacting with TGF-β and its downstream targets [19].

A study from our lab showed that the inhibition of COX-2 by etoricoxib, an NSAID (Nonsteroidal anti-inflammatory drug) specific for the isoform, results in developmental defects in limbs, vascularization, tissue integrity, and the organ patterning of developing chick, where the second most frequently occurring congenital malformations were in the craniofacial region after limb defects [20]. In this study, it was also observed that the selective inhibition of COX-2 reduced only the level of PGE_2_, while the remaining prostanoids maintained their normal titer. Based on these findings and the reported possible interactions of COX-2 with different signaling molecules, we hypothesize that the pathways regulating CNCC delamination and migration might be altered due to COX-2 inhibition, resulting in craniofacial defects in developing embryos. In order to validate this notion, in the present study, the activity of COX-2 was inhibited using etoricoxib, a selective COX-2 inhibitor, and its effect on the regulators of CNCC migration and neural tube closure was ascertained in the chicken embryos. 

## 2. Methodology

### 2.1. Animal Maintenance 

Embryos used for experiments were isolated from eggs of the Rhode Island Red (RIR) breed of *Gallus gallus*. RIR eggs were obtained from the Intensive Poultry Development Block (Vadodara, Gujarat, India). Studies were performed in accordance with the guidelines set by the national regulatory authority for experiments on animals, the CPCSEA, and the protocols were approved by the Institutional Animal Ethics Committee (IAEC; No. MSU-Z/IAEC/09-2020).

### 2.2. Experimental Design 

The eggs were randomly divided into control and treatment groups. Before treatment, air cells were marked by candling the eggs, and thereafter, eggs were wiped with povidone–iodine solution. The precise volume of drug or vehicle was administered into the air cells of the eggs by using an insulin syringe in a sterile laminar flow cabinet. 

Technical-grade etoricoxib (a generous gift from Sun Pharma Advanced Research Company (Vadodara, India)) was used for the dosing of embryos. The solution of etoricoxib was prepared in Milli-Q water by sonication for 2 h at room temperature. Treatment-group eggs were administered with 50 µL of 0.07 mg/mL of etoricoxib. A dose of 0.07 mg/mL (lowest observed adverse effect level) was selected based on a previous dose range study conducted in our lab [20]. The final dose concentration injected in each egg was 3.5 µg (*w/v*) of etoricoxib. Control-group eggs were treated with 50 µL of Milli-Q water.

After dosing, eggs were incubated at 37 ± 0.5 °C and 67 ± 2% relative humidity in a sterile Forma environmental chamber (Thermo Fisher Scientific, Waltham, MA, USA) until Days 1–3 of embryonic development. These embryos were analyzed for mortality and morphological deformities. For all sets of experiments, 30 and 50 eggs, respectively, were used for control and experimental groups to circumvent treatment-induced variance. The Sun–Shepard formula was used to nullify the differences due to the varied sample sizes among groups [21].

### 2.3. Histological Study

Embryo isolation was carried out by the filter ring method. The isolated embryos were rinsed with PBS (Phosphate buffered saline) and fixed with 10% neutral buffered formalin. They were further processed, and paraffin wax blocks of the tissue samples were prepared. Transverse sections of Day-2 embryos were taken using a microtome. These sections were subsequently stained with Harris’ hematoxylin and eosin (Thermo Fisher Scientific, Waltham, MA, USA). Histological details of the prepared samples were visualized using a Leica DM2500 microscope, and images were captured using an EC3 camera (utilizing LAS EZ software, JH Technologies, 213 Hammond Avenue, Fremont, CA, USA).

### 2.4. Western Blot

Total protein was isolated from the head region of the embryos using a lysis buffer containing a protease inhibitor under cold conditions. The Bradford method was used for protein quantification. Proteins were resolved by Polyacrylamide gel electrophoresis (PAGE), consisting of 12% resolving and 4% stacking gels. The resolved proteins were transferred on a Polyvinylidene fluoride (PVDF) membrane by the semidry transfer method. The PVDF membrane was immune-stained for N-cadherin, E-cadherin, vimentin, FoxD3, PCNA, cleaved Caspase-3, and Sox2 by using respective monoclonal antibodies (Sigma Aldrich, St. Louis, MO, USA) diluted in assay buffer in 1:1000 ratios. GAPDH (Glyceraldehyde-6-phosphate dehydrogenase) was used as the internal control for protein levels. Biotinylated secondary antibodies were used to generate colored bands on the membrane.

### 2.5. COX-2 Activity Assay

COX-2 activity was measured by using a kit-based assay (Cayman Chemical, Ann Arbor, MI, USA). The control, treatment, negative control, and positive control were assessed as per the manufacturer’s description. Specific activity was calculated by dividing the total protein values derived from the Bradford assay. Statistical significance of the data was calculated by performing a multiple *t*-test.

### 2.6. RNA Isolation and Quantitative RT-PCR

Total RNA was isolated on Days 1–3 from the head region of embryos using the TRIzol method (Invitrogen, Waltham, MA, USA) and by following the manufacturer’s instructions. The One-Step cDNA Synthesis Kit (Applied Biosystems, Foster City, CA, USA) was used to create cDNA from isolated RNA. Gene amplification reactions were performed using LightCycler96 (Roche Diagnostics, Rotkreuz, Switzerland) and primers for genes WNT3A, TGFB, CDH1, CDH2, VIM, TWIST, MSX1, PCNA, CASP3, and 18S rRNA (Table 1) to identify their relative quantities in the control and treated embryos. The program was set as follows: 3 min at 95 °C for initial melting, 35 cycles (each cycle of 10 s at 95 °C, 10 s at 60 °C, and 10 s at 72 °C), and a final step of 60 s at 65 °C for extra extension. Melting curves for each well were used to confirm the specificity of the products. 18S rRNA was used as an internal loading control. Mean Cq values of the control gene expression were normalized with the internal control gene expression of each group. Fold change in the expression of both genes compared to the control group was calculated using 2^−ΔΔCq^ values as described by Livak and Schmittgen [22]. Data were analyzed by a Student’s *t*-test for significance of the mean difference (GraphPad Software Inc., La Jolla, CA, USA).

## 3. Results

### 3.1. COX-2 Activity

Etoricoxib at a dose of 0.07 mg/mL was administered into the air sacks of fertile eggs on Day-0 and to check its inhibitory effects, and a COX-2 activity assay was performed. A significant decrease in COX-2 activity was recorded in the treated embryos during the early developmental stages, namely Hamburger–Hamilton stage 6 (Day-1), HH 12 (Day-2), and HH 20 (Day-3), known for EMT and CNCC migration, respectively, compared to the control group (Figure 1). Further, it was noticed that COX-2 inhibition led to a marginal yet significant increase in the mortality of embryos at all the stages studied, i.e., Days 1–3 (Table 2). Hence, 50 eggs were incubated for each group to ensure the extraction of 30 live embryos for a given experiment.

### 3.2. Gross Morphology

The sculpting of cranial features coincides with early embryogenesis; hence, the morphology of the cephalic region was studied on Day-1, Day-2, and Day-3 chick embryos. The control-group embryos on Day-1 showed a well-formed neural tube and rhombomeres (Figure 2A), whereas in the treated group, the head fold was disrupted, and the neural tube remained distorted and incompletely formed (Figure 2B–E). Furthermore, on Day-2, embryos of the control group showed defined morphology, with a compact neural tube, perfectly formed rhombomeres, an optic vesicle, and an optic stalk (Figure 3A), while the treated embryos displayed improper closure of the neural tube, a reduced number of somites, and distorted rhombomeres and optic stalks (Figure 3B–D). By the third day, control-group embryos showed a well-structured curved head, perfectly formed optic vesicles, and pharyngeal arches, where neural crest cells further differentiated to form various structures such as frontonasal processes and a mandible (Figure 4A). However, etoricoxib-treated embryos showed a malformed head with no optic vesicle and pharyngeal arches (Figure 4B–D). 

### 3.3. Histological Observations

The extent of hindrance in the migration of cranial neural crest cells upon COX-2 inhibition was studied by the differential staining of Day-2 chick embryos using hematoxylin and eosin stains. The control embryos showed well-formed CNCCs, migrating toward the dorsal side of the neural tube, and a perfectly formed sclerotome (Figure 5A). In contrast, treated embryos showed sparsely distributed CNCCs, indicating delayed formation and migration. The architecture of the sclerotome was also found improper (Figure 5B).

### 3.4. Expression of Genes Involved in EMT and CNCC Migration

As mentioned previously, during CNCC migration, Wnt/β-catenin and TGF-β, along with their downstream molecules, play an important role in EMT, cell survival, and proliferation. Hence, the transcriptional status of WNT3A, TGFB, MSX1, TWIST, CDH1, CDH2, VIM, PCNA, and CASP3 was examined across all three days in the control and treated embryos. On Day-1, the treated embryos elicited a significant reduction in the expression of WNT3A, while TGFB transcripts reduced marginally compared to the control. Meanwhile, MSX1 showed a remarkable ten-fold increase in the treated embryos. TWIST transcripts increased negligibly in the treated embryos, along with a conspicuous increase in the CDH1 gene expression on Day-1 compared to control embryos. A noticeable reduction in CDH2 and VIM gene expression was recorded in COX-2-inhibited embryos on Day-1. PCNA and CASP3 significantly decreased simultaneously (Figure 2F). On Day-2, the etoricoxib-treated embryos showed a remarkable reduction in the expression of the migration of specific genes, namely WNT3A, TGFB, MSX1, TWIST, CDH2, and VIM, whereas CDH1 and CASP3 transcripts increased significantly (Figure 3E). PCNA, a known cell proliferation marker, reduced noticeably on Day-2 compared to the control. Furthermore, on Day-3, the levels of WNT3A, TGFB, VIM, and PCNA transcripts decreased significantly, and MSX1 expression slightly increased in the treated embryos. TWIST showed a marginal elevation at this time, while a significant increase in the levels of CDH1, CDH2, and CASP3 was observed (Figure 4E). 

### 3.5. Levels of Proteins Involved in EMT and CNCC Migration

The protein expression of key regulators of EMT, CNCC proliferation, apoptosis, and migration such as E-cadherin, N-cadherin, FoxD3, vimentin, PCNA, cleaved Caspase-3, and Sox2 was measured by Western blot, followed by a densitometric analysis of the bands. The results revealed a significant upregulation of E-cadherin in the treated embryos compared to the respective control embryos across all three stages studied (Figure 2G, Figure 3F and Figure 4F). On Day-1, N-cadherin reduced significantly in the treated embryos, which also continued for the following stages (Days-2 and 3) compared to the control. FoxD3 protein expression was not detected on Day-1 in both control and treated embryos, while on Day-2, it reduced remarkably in the treatment group. Furthermore, on Day-3, the level of FoxD3 remained low in the treated embryos compared to the control (Figure 3F and Figure 4F). Another crucial EMT regulator, namely vimentin, was not recorded in both control and treated Day-1 embryos, while the Day-2 treated embryos showed no visible alteration compared to the respective control. However, on Day-3, treated embryos expressed a slight reduction in their expression compared to the Day-3 control. Parallel analysis of PCNA revealed that, compared to the control, its expression remained unaltered on Day-1 (Figure 2G) and further plunged significantly at the following stages (Days-2 and 3) (Figure 3F and Figure 4F). Cleaved Caspase 3, on the other hand, reduced marginally on Day 1 in the treated embryos, while its level significantly increased on Day-2 compared to the respective controls. On Day-3, its level elevated negligibly compared to the respective control. Further, Sox2 levels decreased on Day-1 and subsequently on Days-2 and 3 in the treated group of embryos compared to the respective controls. 

## 4. Discussion

Cranial neural crest cells are the most important pool of progenitors and migrate and form various structures of the facial region such as frontonasal, maxillary, and mandibular prominence in developing embryos [23]. In order to understand the mechanisms underlying CNCC migration, which plays a crucial role in the early development of craniofacial structures, the avian model is found to be most suitable. It is closely related to mammals, and the migration pattern of CNCCs is identical and conserved in both these classes of chordates [24]. The factors responsible for CNCC migration are widely accepted to be conserved across various classes of vertebrates. One such factor is COX-2, a member of the cyclooxygenase family involved in cellular processes such as proliferation, migration, angiogenesis, and differentiation [14,15,25]. It is found to be present during the developmental period of the embryo, which regulates the sculpting of various organs [20]. It is conserved among vertebrate species, but its role in embryonic development is still not very well explored. In zebrafish, COX-2-derived PGE_2_ promotes embryonic vasculature maturation [26,27]. These studies substantiate that COX-2 is involved in the governance of cellular and molecular processes required during the early development of embryos. Hence, in the current study, the involvement of COX-2 in craniofacial patterning was investigated by using its specific pharmacological inhibitor, etoricoxib.

The administration of etoricoxib hampered CNCC migration and survival, resulting in several downstream morphological defects such as incomplete head vesicle formation, the absence of an optic vesicle, and defective neural tube formation [28,29,30,31]. As all these are the effects of abnormal CNCC migration or insufficient CNCC survival, the present study focused on the expression patterns of underlying signaling molecules that contribute to craniofacial patterning via CNCC fate determination. A COX-2 activity assay was performed to confirm the inhibition triggered by etoricoxib addition. COX-2 is known to be localized at the region of the neural tube in developing embryos at Days 1–3 [20,32]. In the present results, a reduction in COX-2 activity also caused an alteration in the levels of major regulators of the CNCC migration pathways in the treatment group of embryos. The expression of TGF-β and Wnt3A, along with their downstream signaling factors TWIST, MSX1, Sox2, FoxD3, vimentin, CDH1, and CDH2, was found disturbed under COX-2 inhibition in the current work.

Buch and colleagues illustrated an alteration in Wnt/β-catenin signaling under the inhibition of COX-2, which hampers the regeneration of the lizard tail [33]. The involvement of Wnt3A in the delamination of CNCC has been reported during the early embryogenesis of chicks [34,35]. In the current study, a significant reduction in the expression of WNT3A was noticed upon COX-2 inhibition. Its mRNA level decreased on Day 1 and continued to stoop across Days 2 and 3 in the etoricoxib-treated embryos compared to the respective controls (Figure 2F, Figure 3E and Figure 4E). A decreased WNT3A would have led to a smaller number of CNCCs formed under etoricoxib treatment. Histology results showed that the CNCCs formed were disoriented and lesser in number compared to the control, hence proving the impact of reduced COX-2 activity on normal CNCC formation and migration during early development.

Wnt and TGF-β are known to cooperate in processes involved in EMT [36]. In this case, along with Wnt, TGF-β also showed an unstable trend in Day-1, Day-2, and Day-3 embryos treated with etoricoxib. The gene expression of TGF-β continuously reduced across all stages in the treated embryos. Sela-Donenfeld and Kalcheim reported that an alteration in TGF-β disturbs the migration pattern, as it plays a crucial role in the switching of cadherins during CNCC delamination [37]. MSX1 and TWIST, the downstream mediators of TGF-β, also showed an alteration in gene expression at all stages (Days 1–3) in the etoricoxib treatment group.

Among the downstream regulators of TGF-β, MSX1 controls cellular proliferation and differentiation during early embryonic development. It is highly expressed in CNCCs and plays an important role in regulating the EMT process during embryonic development [38]. In the current results, a significant increase in MSX1 gene expression on Day-1 was followed by a continuous reduction on Day-2 and 3, which substantiates the results obtained through histology. The CNCCs might have delaminated on Day-1 due to an elevation in MSX1 gene expression, although they could not survive or migrate to their destined locations under reduced COX-2 activity. Parallel to MSX1, TWIST gene expression also showed concomitant changes at the initial stages observed here, which further consolidates the idea of cell survival and cell turnover running concurrently to each other [39,40]. In normal conditions, a decline in the level of TWIST causes CNCC to reside in pharyngeal arches and acquire their fates at later stages of migration. In our study, the level of TWIST remained high even at later stages (until Day-3) under the influence of a COX-2 antagonist. This might have hampered the ability of CNCCs to proliferate and differentiate to form a well-organized cranial feature. 

It is well established that epithelial cells undergo an EMT process when E-cadherin is repressed and the cell–cell adhesion is negatively affected. This further leads to cystoskeletal changes in the cells, thus allowing motility [13]. In addition, the loss of N-cadherin is known to cause an interruption in the directed migration phenotype in Xenopus neural crest cells [41]. On the other hand, EMT is a function of well-regulated levels of E-cadherin, N-cadherin, and vimentin [42,43,44]. A decline in CDH1 and an escalation in CDH2 levels lead to the initiation of EMT in the neural tube to form neural crest cells [45]. In the present study compared with respective controls, embryos facing compromised COX-2 activity displayed an increase in E-cadherin both at the gene and protein levels, along with a subsequent reduction in vimentin and N-cadherin on Days-1 and 2, which visibly impeded both CNCC formation and migration. However, by Day-3, both CDH1 and CDH2 increased back to a similar level, which could be due to the compensatory nature of the embryo. Meanwhile, the gene and protein levels of vimentin remained low, even on the third day, indicating the disturbed EMT. This resulted in perturbed patterning of the head region, optic vesicle, and neural tube in the treated embryo.

Furthermore, the expression of FoxD3 was checked, as it is a pivotal marker of CNCCs, facilitating their survival [46,47]. It has been documented that FoxD3 regulates the expression of cell adhesion molecules such as E-cadherin and N-cadherin [48]. In the current study, its protein level expression was recorded from Day-2 onward in both the control and treatment groups. This reinforces the fact that FoxD3 expression only begins after the HH 8 stage in a developing chick embryo [49]. In the treated embryos, FoxD3 protein expression plunged on both Days-2 and 3 compared to the respective controls. This implies the deleterious effects of the drug on the CNCC titer and also its migration.

In order to comment on the cell proliferation in CNCCs, the PCNA transcript and proteins were checked and found to be significantly reduced across all three days in the treated embryos compared to control embryos. In accordance with these data, cleaved Caspase-3 also showed a major upregulation in gene and protein expression at the three stages in the treatment group. These results direct toward the derailed cell proliferation of the delaminated CNCC due to the lacking PCNA status, while an analogous rise in the level of Caspase triggers cell death instead. Overall, CNCCs are unavailable for the normal patterning of frontonasal prominence in the treated embryos due to the deleterious impact of perturbed COX-2 function. A pictorial overview of the result is presented in Figure 6.

Based on our observations, it could be construed that COX-2, perhaps through its downstream effector PGE_2_, regulates the temporal expression pattern of the factors responsible for CNCC formation, proliferation, and migration. Any alteration in the normal titer of COX-2 by accidental prenatal exposure to its commonly used pharmacological inhibitors would result in craniofacial dysmorphism, as observed in the current study due to dysregulation of cranial patterning. 

## Figures and Tables

**Figure 1 jdb-09-00016-f001:**
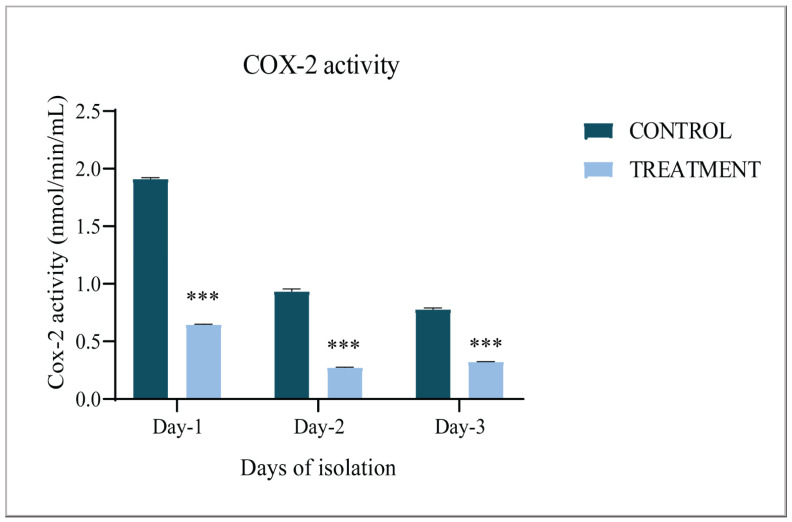
Activity of COX-2 during early development of chick embryos. COX-2 activity in the control and etoricoxib-treated groups of chicks on Days 1–3; *** *p* ≤ 0.001.

**Figure 2 jdb-09-00016-f002:**
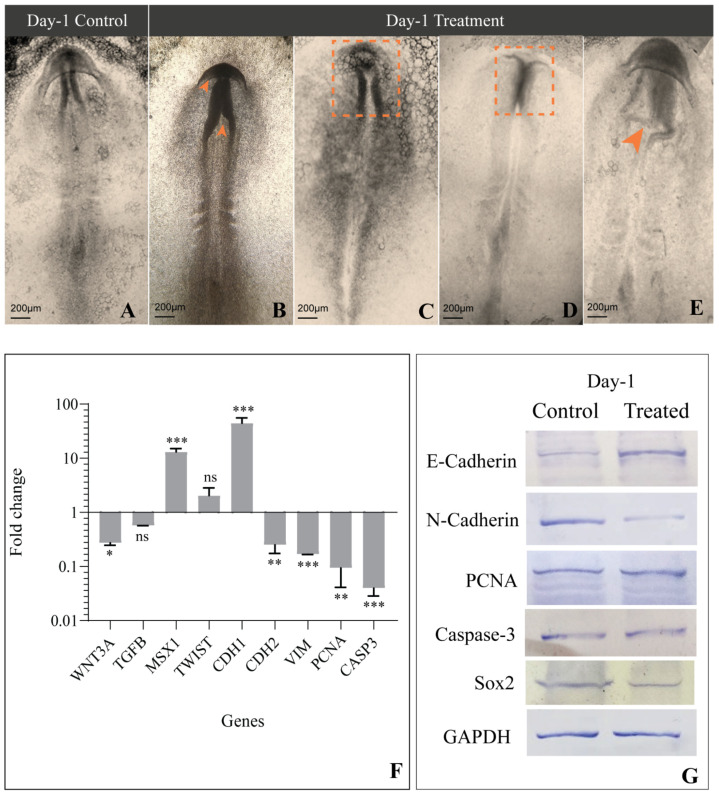
Morphological and molecular aberration due to COX-2 inhibition in the Day-1 embryo. (**A**) Control embryo with a well-developed neural tube, somite, and neural fold meet at the midbrain level; (**B**) Etoricoxib-treated embryos have a defective neural fold formation and meet at the forebrain level (orange arrowheads); (**C**) Etoricoxib-treated embryo showing delayed development and an abnormal neural fold formation (orange dotted square); (**D**) Fusion of neural folds impaired in the etoricoxib-treated embryo (orange dotted square); (**E**) Distorted neural fold marked by an orange arrowhead in the etoricoxib-treated embryo; (**F**) mRNA expression pattern of the genes involved in regulation of neural crest cell migration in etoricoxib-treated embryo. Values are expressed as fold change (mean ± SEM). Fold change values are compared with the control embryo for all genes (ns: non-significant, *: *p* ≤ 0.05, **: *p* ≤ 0.01, ***: *p* ≤ 0.001); (**G**) Western blot image showing the comparative expression of various proteins (E-cadherin, N-cadherin, PCNA, Caspase-3 and Sox2) on Day-1. GAPDH was taken as the loading control (*n* = 3 with 30 eggs per group).

**Figure 3 jdb-09-00016-f003:**
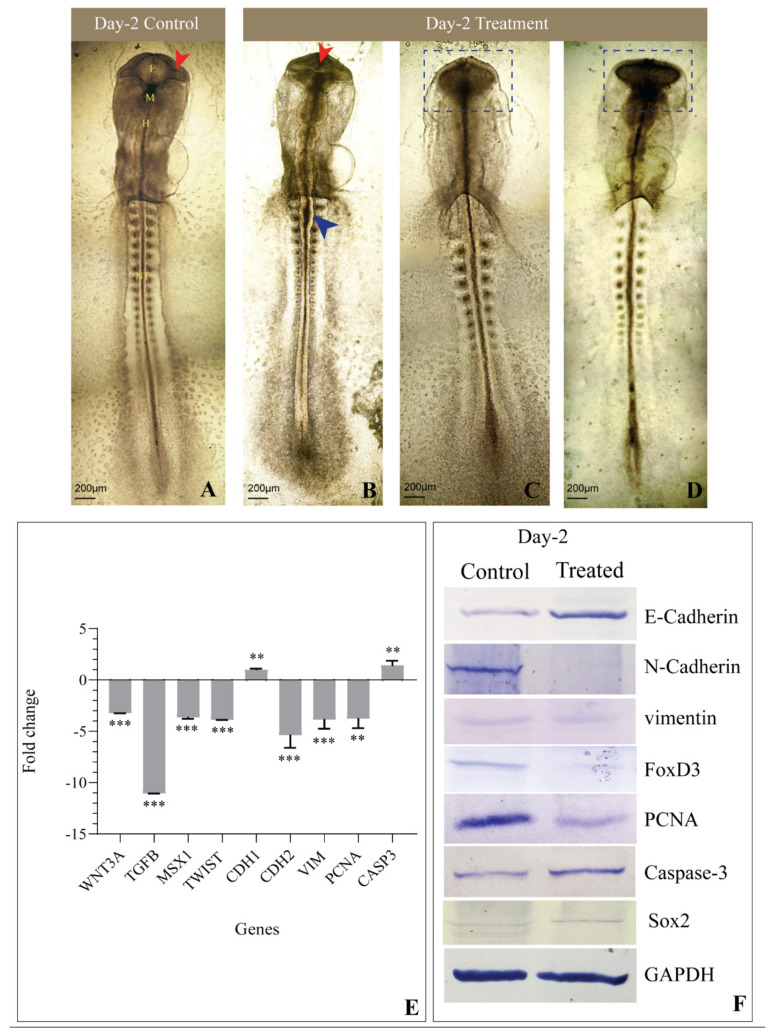
Morphological and molecular changes under COX-2 inhibition in the Day-2 embryo. (**A**) Control embryo with a well-developed primary optical vesicle and optic stalk (red arrowhead) and three primary brain vesicles are clearly visible (where F—forebrain, H—hindbrain, M—midbrain, and NT—neural tube); (**B**) Etoricoxib-treated embryo showing an open neural tube (blue arrowhead), incomplete forebrain and optic vesicle (red arrowhead), and a reduced number of somites; (**C**) The forebrain, midbrain and hindbrain are not formed, with the absence of an optic stalk and an optic vesicle (blue dotted square) in the Day-2 etoricoxib-treated embryo; (**D**) Deformed forebrain region, optic stalk, and optic vesicle are not formed (blue dotted square), and fewer somites are formed in the etoricoxib-treated embryo; (**E**) Transcript-level expression of genes involved in the migration of neural crest cells in etoricoxib-treated Day-2 embryo. Values are expressed as fold change (mean ± SEM). Fold change values of treated embryo are compared with respective controls for all the genes (**: *p* ≤ 0.01, ***: *p* ≤ 0.001); (**F**) Western blot image showing the comparative expression of E-cadherin, N-cadherin, FoxD3, vimentin, PCNA, Caspase-3. and Sox2 on Day-2. GAPDH was taken as the loading control (*n* = 3 with 30 eggs per group).

**Figure 4 jdb-09-00016-f004:**
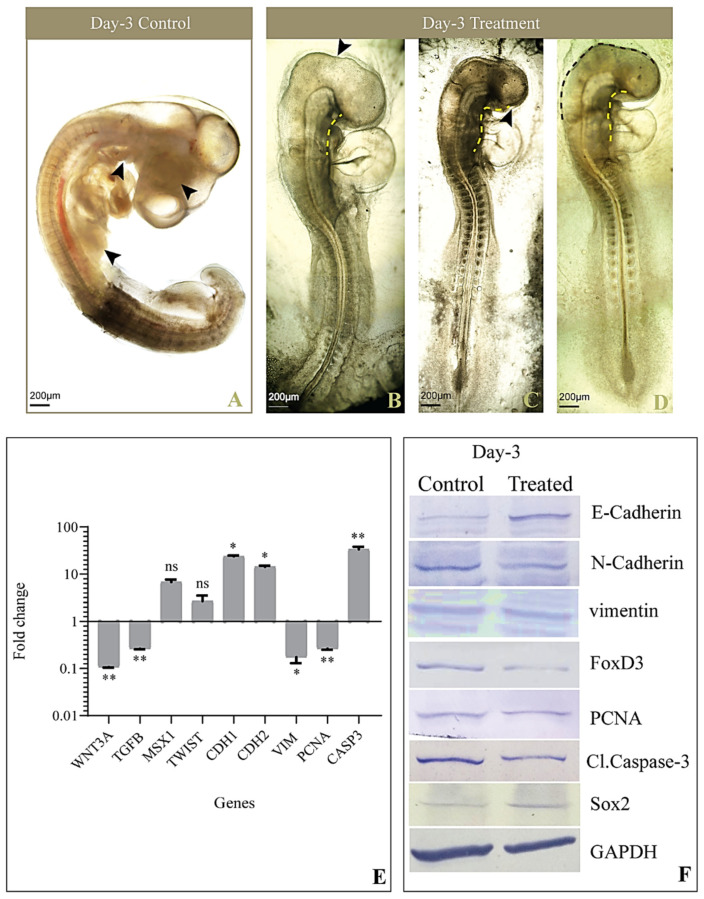
Morphological and molecular observations of Day-3 embryo after treatment. (**A**) Control embryo showing a completely developed forebrain, midbrain and hindbrain, visceral clefts, limb buds, and a primitive eye (black arrowhead); (**B**) Treated embryo indicating poorly developed brain regions (black arrowheads), the absence of a visceral cleft (yellow dotted line), and limb buds; (**C**) Day-3 etoricoxib-treated embryo showing the absence of distinct brain regions, abnormal curving of the body, visceral clefts not formed (yellow dotted line), and the absence of a primitive eye (black arrowhead); (**D**) Etoricoxib-treated embryo showing an abnormal visceral cleft (yellow dotted line), and the head region is defectively formed (brown dotted line); (**E**) Transcript level of genes regulating head formation and neural crest cell migration in the etoricoxib-treated Day-3 embryo. Values are expressed as fold change (mean ± SEM). Fold change values of treated embryo are compared with respective controls for all the genes (ns: non-significant, *: *p* ≤ 0.05, **: *p* ≤ 0.01); (**F**) Western blot image showing the comparative expression of E-cadherin, N-cadherin, FoxD3, vimentin, PCNA, Caspase-3, and Sox2 on Day-3. GAPDH was taken as the loading control (*n* = 3 with 30 eggs per group).

**Figure 5 jdb-09-00016-f005:**
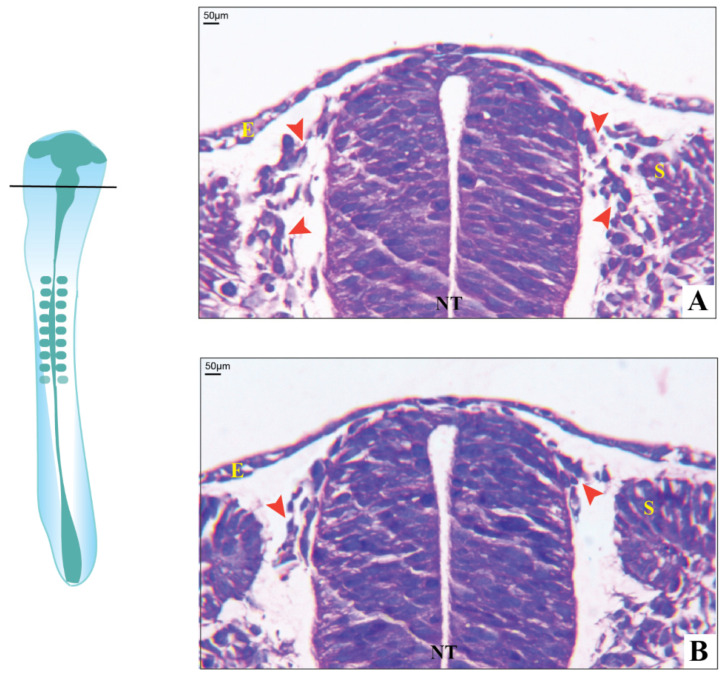
Histology of chick embryos in the transverse section. (**A**) Day-2 control with well-developed neural crest cells migrating towards the ventral side of the neural tube (red arrowheads); (**B**) Day-2 etoricoxib-treated embryo, showing less number of neural crest cells formed along with delayed migration (red arrowheads) (where E—epithelium, NT—neural tube, and S—sclerotome).

**Figure 6 jdb-09-00016-f006:**
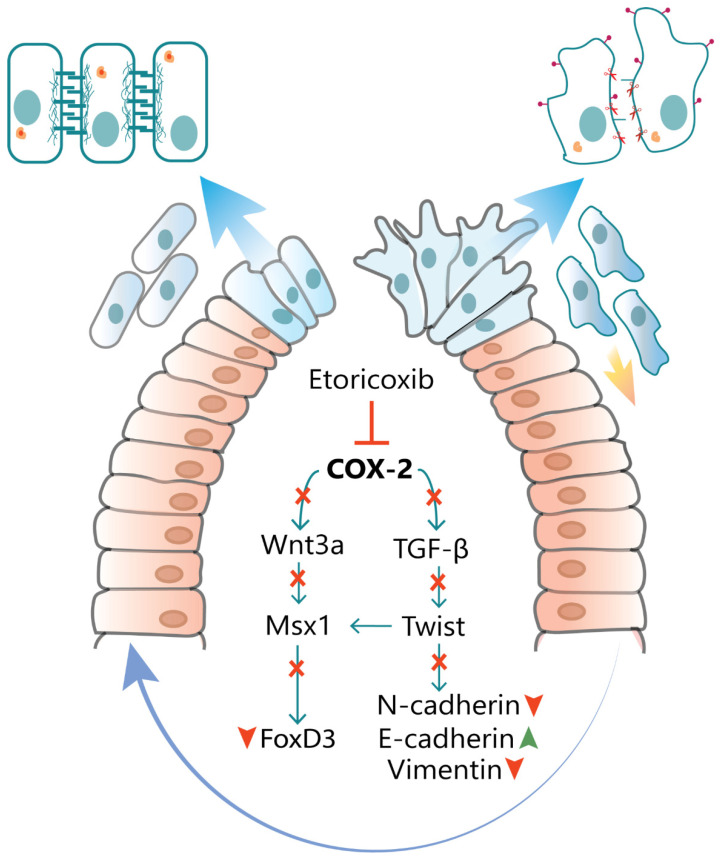
Expression pattern of genes downstream COX-2 upon etoricoxib treatment: a pictorial overview.

**Table 1 jdb-09-00016-t001:** Oligonucleotide primers used for RT-PCR analysis.

Gene Name	Forward Primer (5′-3′)	Reverse Primer (5′-3′)	NCBI Ref ID	Product Length (bp)
WNT3A	TCGGAAACTCCCCTTTCAGC	TGCTCATCTTGCCTGGAG	NM_204675.2	106
TGFB	TCGACTTCCGCAAGGATC	CCCGGGTTGTGTTGGTT	HE646744.1	148
CDH1	GAAGACAGCCAAGGGCCTG	TCTGGTACCCTACCCTCTTG	NM_001039258.2	183
CDH2	AGCCCAGGAGTTTGTAGTG	TTTGGTCCTTTTCTGAGGCCC	XM_025147080.1	114
VIM	GACCAGCTGACCAACGA	GAGGCATTGTCAACATCC	NM_001048076.2	158
TWIST	CGAAGCGTTCACGTCGTTAC	TAGCTGCAATTGGTCCCTCG	NM_204739.2	156
MSX1	CTTACATAGGGCCGAGCCG	CAGGCACAGAACAGATCCCA	NM_205488.2	66
PCNA	TGTTCCTCTCGTTGTGGAGT	TCCCAGTGCAGTTAAGAGCC	NM_204170.2	105
CASP3	AGTCTTTGGCAGGAAAGCCA	CAAGAGTAATAACCAGGAGCG	XM_015276122.2	195
18S rRNA	GGCCGTTCTTAGTTGGTGGA	TCAATCTCGGGTGGCTGAAC	NR_003278.3	144

**Table 2 jdb-09-00016-t002:** Mortality analysis of chick embryos.

Group	Day-1	Day-2	Day-3
Control	2 (1, 4) ***	2 (1, 6) ***	3 (2, 7) ***
Treatment	10 (8, 13)	13 (8, 15)	13 (9, 15)

Mortality observed on Days 1–3 of embryonic development in the control and etoricoxib-treated groups. Values are expressed as mode with the range in parentheses; *** *p* ≤ 0.001.

## Data Availability

The data presented in this study are available within the article.

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
