# Peer review of "Inhibition of Cyclooxygenase-2 Alters Craniofacial Patterning during Early Embryonic Development of Chick"

_jdb, 2021, doi:10.3390/jdb9020016_

Round 1

Reviewer 1 Report

The authors investigate the role of the cyclooxygenase-2 enzyme during craniofacial development of the chick embryo using a chemical inhibitor, characterising the anatomical features of control and treated embryos at three different timepoints and using qPCR and western-blot to investigate the expression level of genes related to cranial neural crest delamination, EMT and migration. The study is well written and clear, however, I think its conclusions could be improved by adding a few experiments and making sure the manuscript is complete before submitting it again:

1) all figure legends were missing. Where are they? While figures are clear they would benefit from legends and annotation, lots of arrows on the embryo pictures in Fig2-3-4 which isn't clear what they indicate. Please re-submit the manuscript including legends, as this is not acceptable without them. 

2)The authors focus on neural crest development but clearly the embryos have other major developmental issues such as completely missing otic vesicles and reduced number of somites. While maybe they are investigating these phenotypes in a separate study (they should!) I feel it would be good to at least discuss them in the discussion section. 

3) the authors chose qPCR and western blots to investigate expression of genes related to neural crest delamination, EMT and migration. While their results are consistent with a neural crest migration phenotype, not a single in situ hybridisation or HNK-1 immunostaining was performed in this study to directly address the distribution of cranial neural crest at different time points studied.

I think the authors should at the bare minimum show in situs for CNC markers at the three time points investigated.

The study would also benefit from using in situs or immunostainings to validate the qPCR /WB findings as the whole head of embryos was used to extract protein/RNA and clearly the whole head is severely affected by cox-2 inhibitors treatment

4) in the final point of the discussion the author mention that the use of cox-2 inhibitors in pregnancy may lead to craniofacial defects. what are the inhibitors used in the clinic and how does the dose used compare with their experimental treatment?

Author Response

Reviewer 1

1) all figure legends were missing. Where are they? While figures are clear they would benefit from legends and annotation, lots of arrows on the embryo pictures in Fig2-3-4 which isn't clear what they indicate. Please re-submit the manuscript including legends, as this is not acceptable without them. 

Response: Even though we had uploaded the figure legends in the portal, due to some technical glitch they did not get incorporated  into the main text of the manuscript that the reviewers received. Presently, we inserted the figure legends into the main text of the revised manuscript, which include annotation of arrows and boxes. The inconvenience caused is deeply regretted.

2)The authors focus on neural crest development but clearly the embryos have other major developmental issues such as completely missing otic vesicles and reduced number of somites. While maybe they are investigating these phenotypes in a separate study (they should!) I feel it would be good to at least discuss them in the discussion section. 

Response: As rightly opined by the reviewer,  craniofacial and other types of developmental anomalies as triggered by COX-2 inhibition, are part of our ongoing studies. However, as suggested, the observed anomalies have been mentioned in the discussion section of the revised manuscript on page number 13.

3) the authors chose qPCR and western blots to investigate expression of genes related to neural crest delamination, EMT and migration. While their results are consistent with a neural crest migration phenotype, not a single in situ hybridisation or HNK-1 immunostaining was performed in this study to directly address the distribution of cranial neural crest at different time points studied.

I think the authors should at the bare minimum show in situs for CNC markers at the three time points investigated.          

The study would also benefit from using in situs or immunostainings to validate the qPCR /WB findings as the whole head of embryos was used to extract protein/RNA and clearly the whole head is severely affected by cox-2 inhibitors treatment

Response: Authors value the suggestion of the reviewer regarding in-situ staining and immunostaining of HNK-1, however procuring the antibody would take more than half a year amid the ongoing pandemic. Therefore, we regret to say that we may not be able to carry out the suggested experiment. However, as an alternate we performed immunostaining of another similar marker FoxD3 in day-1, 2, and 3 embryos of control and treatment groups, while it showed up in visible quantity only in early day-2 stage and the result is added in the revised manuscript as figure 6 on page number 10.

4) in the final point of the discussion the author mention that the use of cox-2 inhibitors in pregnancy may lead to craniofacial defects. what are the inhibitors used in the clinic and how does the dose used compare with their experimental treatment?

Response: Various types of Coxibs (COX inhibitors) are known to be used as pain killers by the pregnant ladies in several countries (Fokunang et al., 2018). Their deleterious effects are also found statistically in cohort study (Dathe et al., 2018). The dosage used in this study causes defects at microgram levels in embryos, which is certainly lesser than the prescribed daily dose in humans. However, the direct correlation may not be possible between chick and humans due to the differences in the type of development in both of these organisms i.e. the amount of the drug that crosses the placental barrier is not clearly documented as of now.

References:

(a) Fokunang, C. N., Fokunang, E. T., Frederick, K., Ngameni, B., & Ngadjui, B. (2018). Overview of non-steroidal anti-inflammatory drugs (nsaids) in resource limited countries. MOJ Toxicol, 4(1), 5-13.

(b) Dathe, K., Padberg, S., Hultzsch, S., Köhler, L. M., Meixner, K., Fietz, A. K., ... & Schaefer, C. (2018). Exposure to COX-2 inhibitors (coxibs) during the first trimester and pregnancy outcome: a prospective observational cohort study. European journal of clinical pharmacology, 74(4), 489-495.

Reviewer 2 Report

In this manuscript, Parmar and colleagues analyse the effects of inhibiting the activity of a molecule, COX-2, on embryo development and neural crest gene expression. They show that COX-2 has a role in the normal expression of various neural crest and related genes, and for normal embryonic neurulation.

Major Comments

  1. There are no figure legends?
  2. There is no Table 2 despite being mentioned on line 143. On a related point, as they mentioned that treatment with the drug caused a significant increase in embryonic lethality, the authors must show that they can rule out that the effects they are observing on gross maldevelopment is not due to the embryo dying due to the treatment applied.
  3. I’m unclear why the title of the paper is the effect on COX-2 on neural crest migration, when all the malformation pictures of embryos show problems related to neural tube closure and other neurulation defects. Likewise, the odd expression patterns of genes like Twist, over the 3 day time course they plotted make it very unclear how COX-2 is actually affecting the neural crest.
  4. The authors should show the expression pattern of Sox2 (e.g. by in situ hybridisation) during normal development at the appropriate stages. This is extremely important to understand the effects of COX-2 inhibition.

Minor Comments

  1. The authors should describe evidence or how they think, from a mechanistic perspective, COX-2 regulates gene expression.

 attached again.

Author Response

Reviewer 2

Major Comments

  • There are no figure legends?

Response: Even though we had uploaded the figure legends in the portal, due to some technical glitch they did not get incorporated  into the main text of the manuscript that the reviewers received. Presently, we inserted the figure legends into the main text of the revised manuscript, which include annotation of arrows and boxes. The inconvenience caused is deeply regretted.

  • There is no Table 2 despite being mentioned on line 143. On a related point, as they mentioned that treatment with the drug caused a significant increase in embryonic lethality, the authors must show that they can rule out that the effects they are observing on gross maldevelopment is not due to the embryo dying due to the treatment applied.

Response: As explained previously, the Table 2 also did not merge with the manuscript as expected. Currently, in the revised manuscript, the table is inserted on page number 4. In order to circumvent the possible statistical error due to marginal rise in mortality induced by the selected dose of the etoricoxib, we have kept more number of eggs in the treatment group so that the number of live embryos in all the groups remain identical. 

  • I’m unclear why the title of the paper is the effect on COX-2 on neural crest migration, when all the malformation pictures of embryos show problems related to neural tube closure and other neurulation defects. Likewise, the odd expression patterns of genes like Twist, over the 3 day time course they plotted make it very unclear how COX-2 is actually affecting the neural crest.

Response: The most frequent malformations, such as absence of optic vesicle, head lob patterning defects, and neural tube closure defects as observed in the treatment group, are all the results of CNCC migration and survival defect. Therefore, the study has rather kept the CNCC migration into focus than the ultimate deformities, by working on the genes related to the CNCC migration and survival. The title of the manuscript already has the word “cranial” neural crest cell migration rather than the neural crest cell migration. The aberrations in gene expression is explained in the discussion however, in the light of reviewers comment few more points are added in revised manuscript to make it explicit.

  • The authors should show the expression pattern of Sox2 (e.g. by in situ hybridisation) during normal development at the appropriate stages. This is extremely important to understand the effects of COX-2 inhibition.

Response: Performing in situ hybridization of Sox-2 at this juncture  is beyond our means. Nonetheless we value the suggestion of the reviewer regarding expression pattern of Sox2 and hence western blot analysis was performed for day-1, day-2 and day-3 and are now incorporated in results as well as in discussion section of the revised manuscript.

Minor Concern

  • The authors should describe evidence or how they think, from a mechanistic perspective, COX-2 regulates gene expression.

Response: Apart from adding notes on how COX-2 act as an upstream regulator of genes involved in CNCC migration, we have now added a summary image (Figure 7 in revised manuscript) depicting the key targets of COX-2 such as Wnt, TGF-β, MSX1, Twist, FoxD3, cadherins as well as vimentin and how the subdued expression of COX-2 (under the influence of etoricoxib) alters their expression resulting in visible morphological defects.

Round 2

Reviewer 1 Report

I thank the authors for including figure legends this time. 

I appreciate their effort to perform an immunostaining for FoxD3. However my comment is that the staining, and especially the control one, is extremely weak and it is difficult to draw meaningful conclusions.Also are the authors sure they are looking for the signal in the right place in the control example? I have the impression there is some more FoxD3 signal rostrally to the magnified inset in Fig 6A-B. While I understand that the current pandemic might limit the ability to procur a new HNK-1 antibody or plasmids for hybridisation probes, I am not sure the FoxD3 results are very conclusive. 

The authors mention that they now discuss the other morphogenetic defects of the embryos in page 13.

However, I have to disagree with their conclusions here: 

"Administration of etoricoxib hampered CNCC migration resulting in several downstream morphological defects such as, incomplete neural tube closure, absence of optic vesicle and reduction in somite numbers. As all these are the effects of abnormal CNCC  migration or insufficient CNCC survival, the present study was focused on the expression patterns of underlying signalling molecules which contribute to craniofacial patterning via CNCC fate determination."

I  seriously doubt that defects in neural tube closure, optic vesicle formation and reduced somite numbers can be downstream effects of defective CNCC migration or survival, as these are independent morphogenetic events. The authors should acquaint themselves better with the literature and give a better explanation than this, also they should support it with appropriate literature citation or what they think is conclusive evidence from their current work. At the moment, I don't see how these other morphogenetic defects might be due to the defects observed in CNCC migration or survival.

Also, their FoxD3 staining does not provide convincing evidence of neural crest migration defects at the moment. 

Finally, I thank the authors for adding comments in the discussion about how their dose of cox2 inhibitor relates to the one used in the clinic. However, I feel the authors should refer to "pregnant women' rather than "ladies" as this is more appropriate given the scientific context. 

Author Response

Reviewer 1

I appreciate their effort to perform an immunostaining for FoxD3. However, my comment is that the staining, and especially the control one, is extremely weak and it is difficult to draw meaningful conclusions. Also are the authors sure they are looking for the signal in the right place in the control example? I have the impression there is some more FoxD3 signal rostrally to the magnified inset in Fig 6A-B. While I understand that the current pandemic might limit the ability to procure a new HNK-1 antibody or plasmids for hybridisation probes, I am not sure the FoxD3 results are very conclusive.

Response: As correctly pointed by the reviewer, authors suspect that the weak signal in control is due to the disperse arrangement of cells as they are migrating during this time point. Whereas, in case of the treated sample, the cells stop migrating and form a ring like structure at the comparable region to the control one, and appear bright due to the clustering at one place. The authors have double-checked that they are looking at the same region as that of control in case of treated sample. Meanwhile, as per the concerns voiced by both reviewers, we understand that the newly added image is not very conclusive. Hence, we are removing the newly added image from the manuscript and updating the second revision accordingly.

However, I have to disagree with their conclusions here:

"Administration of etoricoxib hampered CNCC migration resulting in several downstream morphological defects such as, incomplete neural tube closure, absence of optic vesicle and reduction in somite numbers. As all these are the effects of abnormal CNCC migration or insufficient CNCC survival, the present study was focused on the expression patterns of underlying signalling molecules which contribute to craniofacial patterning via CNCC fate determination."

I seriously doubt that defects in neural tube closure, optic vesicle formation and reduced somite numbers can be downstream effects of defective CNCC migration or survival, as these are independent morphogenetic events. The authors should acquaint themselves better with the literature and give a better explanation than this, also they should support it with appropriate literature citation or what they think is conclusive evidence from their current work. At the moment, I don't see how these other morphogenetic defects might be due to the defects observed in CNCC migration or survival.

Response: Authors have gone through the literature again and we are now adding more references to support our hypothesis about the head and optic vesicle formation being the downstream effects of abnormal CNCC migration and survival. Some of these references are added here as well for the reviewer’s easy comprehension. However, as pointed by the reviewer, not all the defects mentioned in manuscript are downstream to CNCC migration and survival. Authors unanimously decided and corrected the title and content (including conclusion) of the manuscript by adding the phrase “craniofacial development” (Revised title is “Inhibition of cyclooxygenase-2 alters craniofacial patterning during early embryonic development of chick”). CNCC migration has been discussed as a contributory process in the revised manuscript.

  1. Grocott, T., Johnson, S., Bailey, A. P., & Streit, A. (2011). Neural crest cells organize the eye via TGF-β and canonical Wnt signalling. Nature communications, 2(1), 1-6.
  2. Beebe, D. C., & Coats, J. M. (2000). The lens organizes the anterior segment: specification of neural crest cell differentiation in the avian eye. Developmental biology, 220(2), 424-431.

iii. Le Douarin, N. M., Brito, J. M., & Creuzet, S. (2007). Role of the neural crest in face and brain development. Brain research reviews, 55(2), 237-247.

  1. Graham, A., Begbie, J., & McGonnell, I. (2004). Significance of the cranial neural crest. Developmental dynamics: an official publication of the American Association of Anatomists, 229(1), 5-13.

Also, their FoxD3 staining does not provide convincing evidence of neural crest migration defects at the moment.

Response: The FoxD3 staining has been removed from the revised manuscript as it did not seem conclusive to the reviewers. Authors would consider adding immunostaining experiments using various suggested antibodies for the upcoming work and manuscripts. Appreciate your valuable suggestions.

Reviewer 2 Report

The authors have adequately addressed some of my comments, namely the inclusion of the previously missing figure legends and the table, and I appreciate their addition to the discussion on the potential mechanistic targets of COX-2.

However, I continue to disagree with the authors that the various malformations observed by their treatment are as a consequence of failed neural crest migration, as indicated in the text and the title, and defended by the authors in their rebuttal. The morphogenetic defects, including failed neural tube closure, absence of optic vesicle etc are morphogenetic events independent of neural crest migration and their improper development are most certainly not downstream effects of failed neural crest migration. Indeed the authors provide no references that supports their idea, and the literature is overwhelmingly clear on this. As mentioned in my first comments, I believe the authors should change the title and main text as appropriate in this regard. I am also unconvinced by their immunostaining in Fig. 6 which claims to show defective neural crest migration, when the fact that the staining does not appear in a straight line is most likely due to not having a straight neural plate border.

Author Response

Reviewer 2

However, I continue to disagree with the authors that the various malformations observed by their treatment are as a consequence of failed neural crest migration, as indicated in the text and the title, and defended by the authors in their rebuttal. The morphogenetic defects, including failed neural tube closure, absence of optic vesicle etc are morphogenetic events independent of neural crest migration and their improper development are most certainly not downstream effects of failed neural crest migration. Indeed, the authors provide no references that supports their idea, and the literature is overwhelmingly clear on this. As mentioned in my first comments, I believe the authors should change the title and main text as appropriate in this regard.

Response: As recommended by the reviewer, authors have gone through the literature again and we are now adding more references to support our hypothesis about the head and optic vesicle formation being the downstream effects of abnormal CNCC migration and survival. Some of these references are added here as well for the reviewer’s easy comprehension. However, as pointed by the reviewer, not all the defects mentioned in manuscript are downstream to CNCC migration and survival. Authors unanimously agreed to correct the title and content (including conclusion) of the manuscript by adding a phrase “craniofacial development”, keeping CNCC migration as a contributory process. Title has been changed to ‘Inhibition of cyclooxygenase-2 alters craniofacial patterning during early embryonic development of chick.’

  1. i) Grocott, T., Johnson, S., Bailey, A. P., & Streit, A. (2011). Neural crest cells organize the eye via TGF-β and canonical Wnt signalling. Nature communications, 2(1), 1-6.
  2. ii) Beebe, D. C., & Coats, J. M. (2000). The lens organizes the anterior segment: specification of neural crest cell differentiation in the avian eye. Developmental biology, 220(2), 424-431.

iii) Le Douarin, N. M., Brito, J. M., & Creuzet, S. (2007). Role of the neural crest in face and brain development. Brain research reviews, 55(2), 237-247.

  1. iv) Graham, A., Begbie, J., & McGonnell, I. (2004). Significance of the cranial neural crest. Developmental dynamics: an official publication of the American Association of Anatomists, 229(1), 5-13.

I am also unconvinced by their immunostaining in Fig. 6 which claims to show defective neural crest migration, when the fact that the staining does not appear in a straight line is most likely due to not having a straight neural plate border.

Response: The FoxD3 staining has been removed from the manuscript as it did not seem conclusive to the reviewers. Authors would make an effort to consider addition of experiments of immunostaining with various suggested antibodies for the upcoming work and manuscripts. Thank you very much for your critical comments and suggestions.
